# Effect of Iron Complex Source on MWWTP Effluent Treatment by Solar Photo-Fenton: Micropollutant Degradation, Toxicity Removal and Operating Costs

**DOI:** 10.3390/molecules27175521

**Published:** 2022-08-27

**Authors:** Eduardo O. Marson, Ivo A. Ricardo, Cleiseano E. S. Paniagua, Serena M. Malta, Carlos Ueira-Vieira, Maria Clara V. M. Starling, José Antonio Sánchez Pérez, Alam G. Trovó

**Affiliations:** 1Instituto de Química, Universidade Federal de Uberlândia, Uberlândia 38400-902, Brazil; 2Faculdade de Ciências Naturais e Exactas, Universidade Save, Chongoene 0301-01, Mozambique; 3Instituto de Biotecnologia, Universidade Federal de Uberlândia, Uberlândia 38405-319, Brazil; 4Departamento de Engenharia Sanitária e Ambiental, Universidade Federal de Minas Gerais, Belo Horizonte 31270-010, Brazil; 5Solar Energy Research Centre (CIESOL), University of Almería, Ctra. de Sacramento s/n, ES04120 Almería, Spain

**Keywords:** advanced oxidation process, *D. melanogaster*, wastewater

## Abstract

Benzophenone-3, fipronil and propylparaben are micropollutants that are potential threats to ecosystems and have been detected in aquatic environments. However, studies involving the investigation of new technologies aiming at their elimination from these matrices, such as advanced oxidation processes, remain scarce. In this study, different iron complexes (FeCit, FeEDTA, FeEDDS and FeNTA) were evaluated for the degradation of a mixture of these micropollutants (100 µg L^−1^ each) spiked in municipal wastewater treatment plant (MWWTP) effluent at pH 6.9 by solar photo-Fenton. Operational parameters (iron and H_2_O_2_ concentration and Fe/L molar ratio) were optimized for each complex. Degradation efficiencies improved significantly by increasing the concentration of iron complexes (1:1 Fe/L) from 12.5 to 100 µmol L^−1^ for FeEDDS, FeEDTA and FeNTA. The maximum degradation reached with FeCit for all iron concentrations was limited to 30%. Different Fe/L molar ratios were required to maximize the degradation efficiency for each ligand: 1:1 for FeNTA and FeEDTA, 1:3 for FeEDDS and 1:5 for FeCit. Considering the best Fe/L molar ratios, higher degradation rates were reached using 5.9 mmol L^−1^ H_2_O_2_ for FeNTA and FeEDTA compared to 1.5 and 2.9 mmol L^−1^ H_2_O_2_ for FeEDDS and FeCit, respectively. Acute toxicity to Canton S. strain *D. melanogaster* flies reduced significantly after treatment for all iron complexes, indicating the formation of low-toxicity by-products. FeNTA was considered the best iron complex source in terms of the kinetic constant (0.10 > 0.063 > 0.051 > 0.036 min^−1^ for FeCit, FeNTA, FeEDTA and FeEDDS, respectively), organic carbon input and cost-benefit (USD 327 m^−3^ > USD 20 m^−3^ > USD 16 m^−3^ > USD 13 m^−3^ for FeEDDS, FeCit, FeEDTA and FeNTA, respectively) when compared to the other tested complexes.

## 1. Introduction

Studies have demonstrated that conventional municipal wastewater treatment plants (MWWTP) are not designed to remove organic compounds such as pesticides, pharmaceutical drugs and personal care products, among others [1]. These compounds occur in concentrations ranging from ng L^−1^ to µg L^−1^ and are known as micropollutants or contaminants of emerging concern (CEC), as they are potentially harmful to ecosystems and human health [2,3,4,5,6]. Thus, recent studies have evaluated new technologies, i.e., advanced oxidation processes (AOPs) as complementary treatment stages in MWWTP aiming at the degradation of these micropollutants [7]. Although photo-Fenton (Equations (1) and (2)) is an effective process for CEC removal [8,9], one of its main drawbacks is the narrow optimal pH range of operation (pH 2.5–3.0) [10,11], requiring a previous step for acidification followed by neutralization before the discharge of the treated effluent into surface waters, as they usually have neutral pH.
Fe^2+^ + H_2_O_2_ → Fe^3+^ + OH^−^ + HO^•^(1)
[Fe(OH)(H_2_O)_5_]^2+^ + hν → Fe^2+^ + HO^•^(2)

The use of iron-chelating agents such as oxalate (Ox), oxalic acid (OA), tartaric acid (TA), citrate (Cit), nitrilotriacetic acid (NTA), ethylene diaminetetraacetic acid (EDTA) and ethylenediamine-N,N’-disuccinic (EDDS) is one of the strategies used to enable the application of the photo-Fenton process at near-neutral conditions [12,13,14]. A comparison between the use of EDTA, NTA, OA and TA as chelating agents for the degradation of sulfamethoxazole (20 mg L^−1^) in distilled water (DW) at neutral pH and under black-light irradiation (8W, λmax = 365 nm) demonstrated that FeNTA (molar iron/ligand ratio of 1:1.5) appears to be the most appropriate chelating agent. FeNTA is highly biodegradable, adds a lower total organic carbon input compared to other chelating agents and results in low electricity consumption. In contrast, FeOx allows for a higher degradation efficiency (~65%), even in the presence of iron/ligand ratios up to 1:20 [12]. A similar study was performed to compare Ox, Cit, EDDS, EDTA and NTA as Fe-chelating agents for naproxen (200 µg L^−1^) degradation via photo-Fenton in DW at neutral pH and under black-light irradiation (8W, λmax = 365 nm). The best performance was obtained for FeCit at a molar ratio of 1:3 [13].

Micropollutants occur in MWWTP effluents [15,16], and matrix components affect the degradation efficiency and route, influencing the structure and concentration of the transformation products generated during the degradation of micropollutants, chelating agents and the matrix itself [8,17]. However, more studies using real effluents are needed. Only one study has evaluated the efficiency and operating costs associated with the use of different iron sources (EDDS and NTA) on the degradation of a target micropollutant (sulfamethoxazole, 50 µg L^−1^) in MWWTP effluent at neutral pH [18]. In addition, as effluents vary in composition, toxicity must also be used as a response variable for a comprehensive evaluation of the efficiency of each chelating agent in terms of environmental footprint [10].

So far, evaluations on the performance of these chelating agents in DW and MWWTP effluent have been limited to EDDS and NTA [18,19], which are more expensive than EDTA, Cit and Ox [12,13,18]. Therefore, it is critical to perform in-depth studies comparing the efficiencies obtained for the simultaneous degradation of micropollutants and toxicity removal in MWWTP in the presence of each of these agents. These results will allow for the evaluation of process scale-up. In addition, an appropriate comparison must define the best operational conditions for each complex agent (iron concentration, iron/ligand molar ratio and oxidant concentration) in MWWTP to provide deep knowledge about the chemical and biological responses monitored as well as the treatment costs.

In this context, this is the first study focused on the effect of solar photo-Fenton applied in the presence of different chelating agents (Cit, EDTA, EDDS and NTA) for the degradation of micropollutants and toxicity removal at neutral pH in MWWTP effluent under solar irradiation. Two personal care products (benzophenone-3 (BP-3) and propylparaben (PPB)) and the insecticide fipronil (FIP) were selected as model pollutants and were added to the matrix at an initial concentration of 100 µg L^−1^ each. These contaminants were selected because they have been detected in surface water [20,21], raw wastewater [22,23,24,25,26], treated effluent [15,27,28], groundwater [29] and drinking water [30] and are associated with toxic effects [15,31]. Moreover, to the best of our knowledge, no studies evaluating the simultaneous degradation of these compounds by advanced oxidation processes have been reported, and the majority of the reported studies evaluated the degradation of these compounds in distilled water and at higher concentrations (mg L^−1^) [32], which do not represent the reality of the concentration levels (trace level) frequently detected in real matrices.

## 2. Results and Discussion

### 2.1. Influence of the Iron Complex Source, Fe/L Molar Ratios, and H_2_O_2_ Concentration

The source and concentration of the iron complex strongly affected mixture degradation (Figure 1). A large improvement in degradation was observed by increasing the iron complex concentration from 12.5 to 100 µmol L^−1^ for FeNTA, FeEDTA and FeEDDS (Figure 1a–c) due to high concentration of soluble iron in the first 30 min in the presence of these ligands (Appendix A), thus contributing to enhanced Fenton reaction (Equations (1) and (2)). However, this behavior was not observed for FeCit, for which degradation efficiency was limited to 30% for all iron concentrations (Figure 1d). This was justified by the presence of less photoactive iron species. At pH 6.9 (the natural pH of the MWWTP matrix), the predominant iron species are FeOHCit^−^ (90%) and Fe_2_(OH)_2_(Cit)_2_^2–^ (10%), which have lower photoreactivity compared to FeCit [10]. These results are in accordance with the similar H_2_O_2_ consumption observed for the different FeCit concentrations (Appendix A). In addition, poor process efficiency can be attributed to the low stability of the Fe(III)-Cit complex (β = 1.9 × 10^14^) [33] compared to Fe(III)-NTA (β = 7.9 × 10^15^), Fe(III)-EDDS (β = 4.0 × 10^20^) and Fe(III)-EDTA (β = 1.2 × 10^25^) [34]. Although Fe(III)-EDDS presents higher β compared to Fe(III)-NTA, the best degradation results were obtained with FeNTA since the kinetic reaction between EDDS and HO^•^ is five times more efficient than that observed for NTA [35]. This was confirmed by the fast degradation of the target compounds in the first 10 min in the presence of Fe(III)-EDDS, followed by slow degradation until 60–70 min of irradiation (Figure 1c), thus reducing the total dissolved iron concentration (Appendix A) as a consequence of ligand degradation. Thus, an excess of EDDS is needed to guarantee the complex formation and persistence during treatment [14]. Therefore, experiments were performed using the best iron concentration (100 µmol L^−1^) and in the presence of an increased Fe/L molar ratio (Figure 2).

No improvement in the degradation of the target compounds was observed by increasing the Fe/L molar ratio from 1:1 to 1:2 for FeNTA and FeEDTA (Figure 2a,b), as the ratio of 1:1 guaranteed a high concentration of total dissolved iron (Appendix A) during the Fenton reaction. In addition, there was an excess of chelating ligands competing with micropollutants for HO^●^, which decreased the process efficiency [36]. In contrast, the degradation efficiency was enhanced from 62% to 85% by increasing the Fe/L molar ratio from 1:1 to 1:3 for FeEDDS (Figure 2c) and from 1:1 (37%) to 1:5 (84%) for FeCit (Figure 2d), as higher ligand concentrations increased the total dissolved iron availability (Appendix A) and, consequently, H_2_O_2_ consumption (Appendix A). However, no increase was observed by using higher molar ratios of FeEDDS (1:4) and FeCit (1:6) (Figure 2c,d) due to HO^●^ scavenging by the excess of chelating agents [36], as already observed for FeNTA (1:2) and FeEDTA (1:2) (Figure 2a,b). Therefore, 100 μmol L^−1^ of each iron complex and Fe/L molar ratios equivalent to 1:1 (FeNTA and FeEDTA), 1:3 (FeEDDS) and 1:5 (FeCit) were chosen as the best conditions for each ligand. These conditions were then used to evaluate the influence of the H_2_O_2_ concentration (ranged from 0.74 up to 11.8 mmol L^−1^ H_2_O_2_) on the removal of the target CECs (Figure 3).

In the presence of FeNTA, increased concentrations of H_2_O_2_ (0.74 to 11.8 mmol L^−1^) improved the degradation of the target compounds, reaching the LOQ (<4.2 µg L^−1^ for FIP and BP-3 and <1.3 µg L^−1^ for PPB) after 51 and 70 min of reaction with the use of 5.9 and 11.8 mmol L^−1^ H_2_O_2_, respectively (Figure 3a). However, H_2_O_2_ concentrations above 5.9 and up to 11.8 mmol L^−1^ required higher treatment times to reach the LOQ (<4.2 µg L^−1^ for FIP and BP-3 and <1.3 µg L^−1^ for PPB) (Figure 3a) by 37% due to inefficient parallel reactions (Equation (3)).

Similar behavior was observed for FeEDTA (Figure 3b), for which 59% degradation (34 min) was obtained in the presence of 0.74 mmol L^−1^, while 88% (39 min) removal was achieved in the presence of 5.9 mmol L^−1^ of H_2_O_2_. The previously mentioned conditions reached levels below the LOQ (<4.16 µg L^−1^ for FIP and BP-3 and <1.33 µg L^−1^ for PPB) after 65 min. These results are associated with higher generation of HO^●^ radicals in the presence of higher H_2_O_2_ concentrations due to reaction with ferrous ions. However, when 11.8 mmol L^−1^ H_2_O_2_ was added to the system containing FeEDTA, there was a decrease in process efficiency, which reached a maximum of 84% degradation within 51 min (Figure 3b) as a consequence of inefficient parallel reactions between excess H_2_O_2_ and HO^●^ (Equation (3)) and other inefficient reactions driven by secondary radicals (Equations (4)–(12)), thus hindering the degradation of the target compounds [13,37].
H_2_O_2(aq)_ + HO^•^_(aq)_ → HO_2_^•^_(aq)_ + H_2_O_(l)_ → k = 2.7 × 10^7^ L mol^−1^ s^−1^(3)
HO_2_^•^_(aq)_ + HO^•^_(aq)_ → H_2_O_(l)_ + O_2(g)_ → k = 7.1 × 10^9^ L mol^−1^ s^−1^(4)
HO_2_^•^_(aq)_ + H_2_O_2(aq)_ → O_2(g)_ + H_2_O_(l)_ + HO^•^_(aq)_ → k = 0.5 L mol^−1^ s^−1^(5)
HO_2_^•^_(aq)_ + HO_2_^•^_(aq)_ → O_2(g)_ + H_2_O_2(aq)_ → k = 8.23 ± 0.7 × 10^5^ L mol^−1^ s^−1^(6)
HO_2_^●^_(aq)_ ⇄ O_2_^●–^_(aq)_ + H+_(aq)_ pK_a_ = 4.8(7)
HO_2_^●^_(aq)_ + O_2_^●–^_(aq)_+ H_2_O_(l)_ → H_2_O_2(aq)_ + O_2(g)_ + HO^−^_(aq)_(8)
HO_2_^−^_(aq)_ + HO^•^_(aq)_ → HO_2_^•^_(aq)_ + OH^−^_(aq)_ → k = 7.5 × 10^9^ L mol^−1^ s^−1^(9)
O_2_^•^^−^_(aq)_ + H_2_O_2(aq)_ → O_2(g)_ + OH^−^_(aq)_ + HO^•^_(aq)_ → k = 0.13 L mol^−1^ s^−1^(10)
O_2_^•^^−^_(aq)_ + HO^•^_(aq)_ → H_2_O_(l)_ + O_2(g)_ → k = (0.66–1.4) × 10^10^ L mol^−1^ s^−1^(11)
HO^•^_(aq)_ + HO^•^_(aq)_ → H_2_O_2(aq)_ k = (5–8) × 10^9^ L mol^−1^ s^−1^(12)

Regarding the use of FeEDDS, increasing the H_2_O_2_ concentration from 0.74 mmol L^−1^ to 1.5 mmol L^−1^ led to better degradation of the target compounds, reaching 66% and 88% after 34 min and 99 min, respectively. However, no improvement in degradation was observed in the presence of higher H_2_O_2_ concentrations (between 2.9 and 11.8 mmol L^−1^) (Figure 3c), indicating the occurrence of inefficient parallel reactions [13]. Therefore, 1.5 mmol L^−1^ H_2_O_2_ was selected as the best oxidant concentration for FeEDDS. This concentration is equivalent to 25% of that required for FeNTA and FeEDTA (Figure 3a,b). The lower H_2_O_2_ concentration required in the presence of FeEDDS is due to the generation of HO^•^ radicals from EDDS^•^ due to the fast photolysis of the FeEDDS complex (Equations (1) and (13)–(17)) [38]. This explains the results obtained during the irradiation of FeEDDS in the absence of H_2_O_2_ (control experiment), which reached 40% degradation after 60 min of irradiation (Appendix A), while less than 10% degradation were observed by direct photolysis and Fenton-like processes (FeEDDS/H_2_O_2_) (Appendix A).
Fe^3+^–EDDS_(aq)_ + hν → [Fe^3+^–EDDS]^*^_(aq)_ → Fe^2+^_(aq)_ + EDDS^●^_(aq)_(13)
EDDS^●^_(aq)_ + O_2(g)_ → O_2_^●–^_(aq)_ + Products_(aq)_(14)
Fe^3+^_(aq)_ + O_2_^●–^_(aq)_ → Fe^2+^_(aq)_ + O_2(g)_(15)
Fe^2+^_(aq)_ + O_2_^●–^_(aq)_ + 2H_2_O_(l)_ → Fe^3+^_(aq)_ + H_2_O_2(aq)_ + 2HO^−^_(aq)_(16)
EDDS^●3–^_(aq)_ + HO^−^_(aq)_ → EDDS^4–^_(aq)_ + HO^●^_(aq)_(17)

Regarding FeCit, increasing H_2_O_2_ concentrations from 0.74 to 2.9 mmol L^−1^ improved degradation rates, reaching the LOQ (<4.2 µg L^−1^ for FIP and BP-3 and <1.3 µg L^−1^ for PPB) after 19 min of reaction (Figure 3d). On the other hand, further increases in the H_2_O_2_ concentration from 2.9 to 5.9 and 11.8 mmol L^−1^ resulted in a decrease in the degradation efficiency (Figure 3d), as reported for other ligands due to the occurrence of inefficient parallel reactions (Equations (3)–(12)) [13,37]. In addition, similar to the observations made for FeEDDS, a lower H_2_O_2_ concentration of FeCit was needed when compared to FeNTA and FeEDTA since the photolysis of the predominant iron species (FeOHCit^−^, 90%) at pH 6.9 (the natural pH of the MWWTP matrix) produces H_2_O_2_ (Equations (6)–(8), (18) and (19)), thus requiring a lower concentration of this reagent [39]:[FeOHCit]^−^_(aq)_ + hν → Fe^2+^_(aq)_ + 3–HGA^2●−^_(aq)_ → k = 1.3 × 10^−2^ M^−1^ s^−1^(18)
3–HGA^2●−^_(aq)_ + O_2(g)_ → 3–HGA^2−^_(aq)_ + CO_2(g)_ + O_2_^●−^_(aq)_ → k = 1.0 × 10^6^ M^−1^ s^−1^(19)

Furthermore, the efficiency of the solar photo-Fenton process for the treatment of real MWWTP effluents containing BP-3, FIP and PPB must not be evaluated only with regard to kinetic aspects since other variables (acute toxicity after the treatment and the costs of the reagents) are critical to defining the best iron complex to enable the application of the photo-Fenton process at a real scale.

### 2.2. Acute Toxicity Assays: Drosophila Melanogaster Lifespan

MWWTP effluent containing CEC was highly toxic to *D. melanogaster* since almost all flies were dead after 7 days (Figure 4 and Appendix A). This effect may be attributed to FIP, as no flies survived after 7 days of exposure to MWWTP spiked exclusively with FIP (Appendix A). In contrast, exposure of test organisms to MWWTP effluent spiked exclusively with BP-3 or PPB resulted in 70% and 64% survival, respectively, after 15 days of exposure (Appendix A).

Samples collected after 20 min of the photo-Fenton process in the presence of FeEDDS and FeCit systems showed a considerable percentage of fly survival (35% and 50% for FeEDDS and FeCit, respectively) after 15 days of exposure. These results suggests that the by-products generated during these treatments were less toxic than the parent compounds (Figure 4c,d). An increased reaction time (60 min) showed no improvement in the percentage of fly survival (39% and 56% survival after 15 days of exposure to FeEDDS and FeCit, respectively). These results were expected since no increase in target-compound degradation efficiency was observed between 20 min (t_30W_ = 33 min for FeEDDS and t_30W_ = 19 min for FeCit) and 60 min (t_30W_ = 99 min for FeEDDS and t_30W_ = 60 min for FeCit) of treatment carried out in the presence of FeEDDS and FeCit (Figure 3c,d). The higher fly mortality observed for the system containing FeEDDS when compared to the FeCit system is due to a lower FIP degradation rate in the presence of FeEDDS (87%) when compared to FeCit (<LOQ = 4.2 µg L^−1^) (Figure 3c,d), as FIP is the main compound responsible for the toxic effect upon *D. melanogaster* (Appendix A).

A different behavior concerning toxicity was observed for samples obtained in the presence of FeEDTA and FeNTA. For FeEDTA, the sample obtained after 20 min (t_30W_ = 26 min) of photo-Fenton killed 85% of the flies after 15 days (Figure 4b), which can be attributed to the residual FIP concentration (46 µg L^−1^), as only a 54% FIP degradation was achieved within this treatment time (Figure 3b). As previously discussed for FeEDTA, the quantification limits (<4.2 µg L^−1^ for FIP and BP-3 and <1.3 µg L^−1^ for PPB) of the target compounds were only reached after 60 min (t_30W_ = 77 min). Hence, the sample obtained after 60 min of solar photo-Fenton using FeEDTA provided higher survival of *D. melanogaster* (53% survival after 15 days) (Figure 4b). Concerning FeNTA, samples obtained after 20 min (t_30W_ = 13 min) of solar photo-Fenton treatment reached 95% fly mortality (Figure 4a), which can also be attributed to incomplete FIP degradation (78%) (22.0 µg L^−1^ residual concentration) (Figure 4a). Samples obtained after 60 min (t_30W_ = 51 min) reached the LOQ (<4.2 µg L^−1^ for FIP and BP-3 and <1.3 µg L^−1^ for PPB) (Figure 3a) and resulted in 54% survival of flies after 15 days of exposure (Figure 4a).

The exposure of flies to controls containing H_2_O and MWWTP effluent without CECs showed 90% and 80% fly survival after 15 days of exposure (Appendix A). In addition, control experiments performed to assess effluent degradation in the absence of CECs using each of the iron complexes (Appendix A) resulted in 70–80% fly survival after 15 days, indicating that nontoxic transformation products were generated from MWWTP effluent and iron complexes. These results suggest that the residual FIP concentration (<4.2 µg L^−1^) implied toxicity to test organisms, even after the LOQ is reached (<4.2 µg L^−1^ for FIP and BP-3 and <1.3 µg L^−1^ for PPB), resulting in nearly 50% fly survival after 15 days (Figure 4).

### 2.3. Cost Assessment

Ligand and oxidant costs were calculated for the treatment of 1 m^3^ of effluent from MWWTP, and values were obtained considering the reagent concentrations that led to the best performance and the commercial price for these reagents (Appendix A). One of the advantages of solar photo-Fenton is the use of sunlight as the radiation source, which is economically attractive since the costs of artificial radiation are waived [13]. In this work, electrical costs were not considered in the calculations, and all ligands were quoted at Sigma-Aldrich for better comparison between them, as it is the single seller of EDDS. Even so, the prices from other companies may be different from those presented in this work.

Considering that this is a comparative study, the prices revealed in Table 1 are estimates and are presented in a relative form by comparison with the lower price (price equals 1.0). The costs associated with iron concentration were the same for all iron complexes (Table 1) since the best iron concentration was the same for all tested complexes (100 µmol L^−1^). The treatment performed in the presence of FeEDDS showed the lowest oxidant cost, yet it was the most expensive process because EDDS is the most expensive ligand (USD 2984 kg^−1^) (Appendix A). Solar photo-Fenton applied in the presence of FeEDDS was estimated to be 71-fold more costly than the same treatment performed with the NTA ligand (Table 1). Although citrate is the cheapest complex (USD 86 kg^−1^) (Appendix A), the costs with this ligand were the highest, as a higher Fe:Cit molar ratio (1:5) is required to reach the LOQ. The costs associated with FeCit were 2.9-fold higher than those calculated for NTA (Table 1). FeEDTA and FeNTA showed the same costs associated with the oxidant, which was 3.9 higher than the oxidant price for EDDS. In contrast, the process using FeNTA was cheaper than using FeEDTA (Table 1) due to the lower cost of NTA (USD 173 kg^−1^) when compared to EDTA (USD 206 kg^−1^) (Appendix A).

Thus, the costs associated with the processes evaluated in this study follow the ascending order: FeNTA < FeEDTA < FeCit < FeEDDS. Kinetic constants obtained for the photo-Fenton process (k) were calculated considering a pseudo-first-order model for the degradation of the target compounds. High values of the coefficient of determination (R^2^) were obtained, indicating that all ligands respect the pseudo-first-order model (Table 1). Kinetic constants also indicate the crescent order: FeEDDS < FeEDTA < FeNTA < FeCit. In other words, FeCit and FeNTA showed faster degradation rates and lower half-life times (t_1/2_) when compared to FeEDTA and FeEDDS. Moreover, the addition of ligands to the effluent added total organic carbon (TOC) to the MWWTP matrix, which was equivalent to 36 mg L^−1^ for FeCit and FeEDDS, 12 mg L^−1^ for FeEDTA and 7 mg L^−1^ for FeNTA. As the original effluent TOC was 38 mg L^−1^ (Appendix A), Cit and EDDS represented a TOC increase of 95% each, while EDTA and NTA contributed 32% and 7%, respectively. Although effluent TOC increased, the use of these ligands enabled treatment conduction at near-neutral pH, which is a drawback of the conventional photo-Fenton process, and improved CEC degradation (treatment in the absence of complexes achieved only 17% degradation efficiency) (Appendix A). However, the choice of the best ligand must also consider the kinetic constant, added TOC content and lower costs. Therefore, although FeCit showed the highest kinetic constant, the higher Fe/L molar ratio (1:5) associated with this ligand implied in a higher TOC increase and higher costs.

Thus, considering the kinetic parameters, TOC increase, total process costs and toxicity to *D. melanogaster*, FeNTA presented the most cost-efficient alternative for the treatment of CEC in MWWTP effluent at near-neutral pH using solar photo-Fenton. However, it is important to mention that this work was performed in lab-scale experiments. Therefore, an experiment at a larger scale must be conducted to confirm these results and verify the economic feasibility of applying these processes. Soriano-Molina et al. [18] investigated sulfamethoxazole removal in actual MWWTP effluents using FeNTA and FeEDDS for continuous flow solar photo-Fenton and also concluded that FeNTA was the most cost-effective ligand to be used in the solar photo-Fenton process, even when compared to classic photo-Fenton, as it showed higher costs associated to the iron source (FeSO_4_·7H_2_O) and acidification (6% of the total treatment cost), thus resulting in less efficiency associated to higher costs (30% more expensive) when compared to FeNTA.

## 3. Materials and Methods

### 3.1. Reagents

All chemical solutions were prepared in ultrapure water (18.2 MΩ cm) produced by a Milli-Q water purification system. FIP (98.77 wt%) was acquired from Sinochem Ningbo (Lian Yun Gang City, Jiangsu Province, China). PPB (C_10_H_12_O_3_, 99 wt%), BP-3 (C_14_H_12_O_3_, 98 wt%), NTA (C_6_H_6_NNa_3_O_6_), EDDS trisodium salt solution (C_10_H_13_N_2_Na_3_O_8_) and potassiumtitanium oxide oxalate (C_4_K_2_O_9_Ti·2H_2_O, ≥90 wt%) were purchased from Sigma-Aldrich (São Paulo, Brazil). Ferric nitrate nonahydrate (Fe(NO_3_)_3_·9H_2_O), ferrous sulfate heptahydrate (FeSO_4_·7H_2_O), EDTA disodium salt dihydrate (C_10_H_14_N_2_O_8_Na_2_·2H_2_O), sodium citrate dihydrate (Na_3_C_6_H_5_O_7_·2H_2_O), hydrogen peroxide (H_2_O_2_) (30 wt%), sodium sulfite (Na_2_SO_3_) and 1.10-phenanthroline (C_12_H_8_N_2_·H_2_O) were provided by Synth (Diadema, Brazil). Hydroxylamine hydrochloride (NH_2_OH·HCl) was obtained from Êxodo (Sumaré, Brazil), HPLC-grade methanol (CH_3_OH) was obtained from J. T. Baker (Aparecida de Goiânia, Brazil), acetic acid (CH_3_COOH) was obtained from Panreac (Barcelona, Spain) and anhydrous sodium acetate (CH_3_COONa) and sulfuric acid (H_2_SO_4_) were both furnished by Dinâmica (Indaiatuba, Brazil).

### 3.2. MWWTP Effluent

MWWTP effluent samples were collected at the Municipal Department of Water and Sewage located in Uberlândia, Brazil (18° 55′ 08″ S, 48° 16′ 37″ W). Sampling was carried out in the flotation channels after the following stages: preliminary treatment (bar screens and desanders), upflow anaerobic sludge blanket (UASB), coagulation–flocculation (ferric chloride, FeCl_3_) and flotation. A physicochemical characterization of the effluent matrix was carried out after sampling (Appendix A), and samples were stored in amber bottles and kept under refrigeration at 4 °C before and during the degradation experiments.

### 3.3. Photodegradation Experiments

All photodegradation experiments were conducted in a dark glass vessel (5.5 cm depth, 0.13 cm width and 5.0 cm of optical path) under magnetic stirring. A 500 mL volume of the MWWTP effluent was spiked with a mixture containing target compounds (500 mg L^−1^ of each CEC prepared in acetonitrile) in order to obtain a final concentration of 100 µg L^−1^ of each CEC.

Experiments were carried out under solar radiation between 10 a.m. and 2 p.m. An aliquot was taken before treatment kick-off (–10 min) for HPLC analysis. Iron-complex solutions were prepared in the dark at the desirable iron/ligand ratios and apart from the mixture by mixing the organic ligand to Fe^3+^ solutions at the appropriate concentration. In the sequence, solutions were stirred for 5 min in order to ensure the formation of iron complexes [40]. Subsequently, the iron complex solution was added to the mixture. After five minutes of stirring in the dark, another sample was collected (–5 min) for chromatographic analysis and total iron quantification. The oxidant was then added to the solution, and after five minutes of stirring (Fenton process, 0 min), samples were taken for chromatographic analysis and for the quantification of total iron and H_2_O_2_ concentrations. From this moment on, the effluent was exposed to solar irradiation (indicated in Figure 1, Figure 2 and Figure 3 by RAD-ON), and aliquots were sampled for the previously mentioned analyses at pre-established times (5, 10, 20, 30, 40, 50 and 60 min). Incident irradiation (W m^−2^) was monitored by a radiometer (MU-100) containing an ultraviolet sensor (250–400 nm), and registered data were used to calculate the normalized irradiation time (t_30W_) [41].

Three sets of experiments were performed at an initial pH of 6.9 (the natural pH of the MWWTP matrix) in order to determine the best operational parameter conditions for each ligand: (i) the effect of iron concentration (12.5, 25, 50 and 100 µmol L^−1^) at a fixed Fe/Ligand (Fe/L) molar ratio (1:1) and a fixed initial concentration of 5.9 mmol L^−1^ H_2_O_2_ (200 mg L^−1^ H_2_O_2_) [13]; (ii) the effect of the Fe/L molar ratio (1:2 for FeCit, FeEDDS, FeEDTA and FeNTA; 1:3 and 1:4 for FeCit and FeEDDS; and 1:5 and 1:6 for FeCit) using the best iron concentration (100 µmol L^−1^) and 5.9 mmol L^−1^ H_2_O_2_; (iii) the influence of the H_2_O_2_ concentration (0.74, 1.5, 2.9, 5.9 and 11.8 mmol L^−1^ H_2_O_2_, equivalent to 25, 50, 100 and 400 mg L^−1^, respectively) under the best iron concentration (100 µmol L^−1^) and molar ratios (1:1 for FeEDTA and FeNTA, 1:3 for FeEDDS and 1:5 for FeCit).

Control experiments using the best Fe/L molar ratios and iron and H_2_O_2_ concentrations were also performed under solar irradiation (photolysis, Fe/L and H_2_O_2_) and in the dark (H_2_O_2_ and FeL/H_2_O_2_) at pH 6.9 (the natural pH of the MWWTP matrix). All experiments were performed in triplicate, and the figures show average values with standard deviation error bars.

### 3.4. Chemical and Bioassay Analysis

An LC-6AD high-performance liquid chromatographer (Shimadzu) equipped with a model SPD-M20A (Shimadzu) UV-DAD detector and a Phenomenex C-18 column (Luna 5 μm, 250 × 4.6 mm) was used to monitor the concentration of each CEC. Isocratic elution was applied using a mobile phase composed of 75% acetic acid (0.01% *v/v*) in Milli-Q water and 25% methanol. The flow rate was 1 mL min^−1^. Retention times of 5.6 ± 0.1 (PPB), 9.0 ± 0.1 (FIP) and 10.2 ± 0.1 (BP-3) min were obtained. The limits of quantification (LOQs) were: 4.2 µg L^−1^ for FIP (279 nm) and BP-3 (289 nm) and 1.3 µg L^−1^ for PPB (256 nm), thus allowing for the assessment of up to 96–99% degradation.

The average degradation of the mixture (*C*/*C*_0_) was evaluated by calculating the difference between the average of the initial concentrations and the average of the concentrations after a certain reaction time (t_30W_) for the sum of the target compounds (Equation (20)):(20)$$C/C_{0}=\frac{\sum C_{0}-\sum C_{t_{30}Wi}}{\sum C_{0}}$$
where $C_{0}$ is the initial concentration of each target compound, *wi* is the % in weight of each contaminant and $C_{t_{30W}}$ is the concentration of each target compound at t_30W_.

The concentration of H_2_O_2_ was determined by the spectrophotometric method using titanium oxalate [42], and the total dissolved iron was measured by Standard Method 3500-Fe B [43].

Residual H_2_O_2_ was removed from aliquots prior to the HPLC analysis by adding excess Na_2_SO_3_ (1 g L^−1^). In contrast, excess H_2_O_2_ present in the samples submitted to toxicity assays was removed by the addition of bovine catalase (2 g L^−1^) prior to sample filtration and exposure to test organisms [44].

The acute toxicity of treated and untreated samples towards *Drosophila melanogaster (D. melanogaster)* was estimated according to the methodology described by Gomes Júnior et al. [44] and Gonçalves et al. [45]. Further details are provided in the Appendix A section. All *D. Melanogaster* bioassays were also performed in triplicate.

## 4. Conclusions

The efficiency of solar photo-Fenton using four iron ligands was assessed for the simultaneous degradation of benzophenone-3, fipronil and propylparaben in MWWTP effluent for the first time. Treatment systems were compared in terms of kinetics, toxicity and costs. Although all iron complexes were efficient for the degradation of benzofenone-3, fipronil and propylparaben, the best operational conditions for each system were different. In the presence of 100 µmol L^−1^ of ferric ions, the best Fe/L molar ratios and oxidant concentrations were 1:1 and 5.9 mmol L^−1^ H_2_O_2_ for FeEDTA and FeNTA; 1:3 and 1.5 mmol L^−1^ H_2_O_2_ for FeEDDS; and 1:5 and 2.9 mmol L^−1^ H_2_O_2_ FeCit. These results reveal the need to optimize the treatment conditions of each system since distinct iron complex species associated with different photo-activity and stability are present in each system and impact the degradation efficiency. In addition, acute toxicity to Canton S. strain *D. melanogaster* flies reduced significantly after treatment for all evaluated iron complexes, which indicates the formation of low-toxicity transformation products in the process. Considering the results obtained for the simultaneous degradation of benzofenone-3, fipronil and propylparaben in MWWTP effluent, toxicity and costs, FeNTA is presented as the best option for the application of solar photo-Fenton at neutral pH. Thus, the results obtained in this study reveal that solar photo-Fenton modified with organic iron complexes may be a viable alternative as a post-treatment of conventional MWWTP aiming at CEC and toxicity removal.

## Figures and Tables

**Figure 1 molecules-27-05521-f001:**
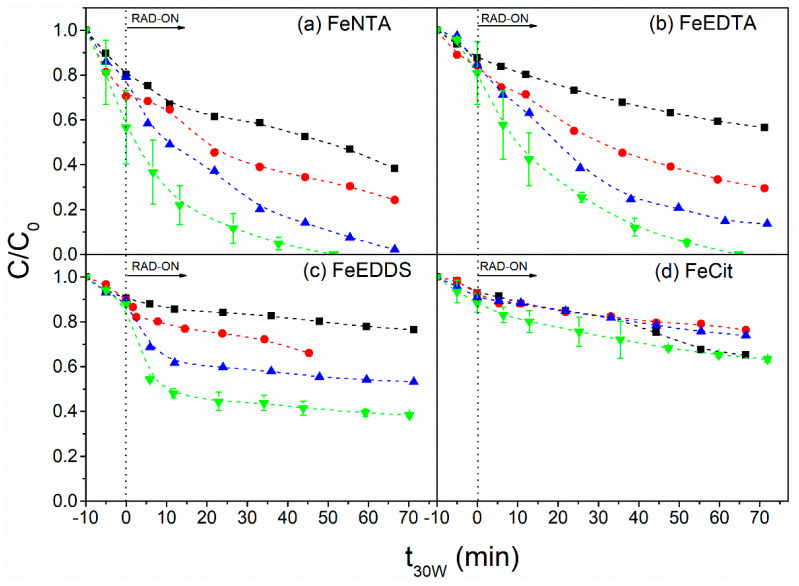
Influence of iron complex source and concentration (■ 12.5 µmol L^−1^, ● 25 µmol L^−1^, ▲ 50 µmol L^−1^ and ▼ 100 µmol L^−1^) on the degradation of the mixture of benzophenone-3, fipronil and propylparaben using the iron complexes (**a**) FeNTA, (**b**) FeEDTA, (**c**) FeEDDS and (**d**) FeCit. Initial conditions: [CEC] = 100 µg L^−1^ (for each compound), [H_2_O_2_] = 5.9 mmol L^−1^, Fe/L = 1:1; pH = 6.9 (the natural pH of the MWWTP effluent).

**Figure 2 molecules-27-05521-f002:**
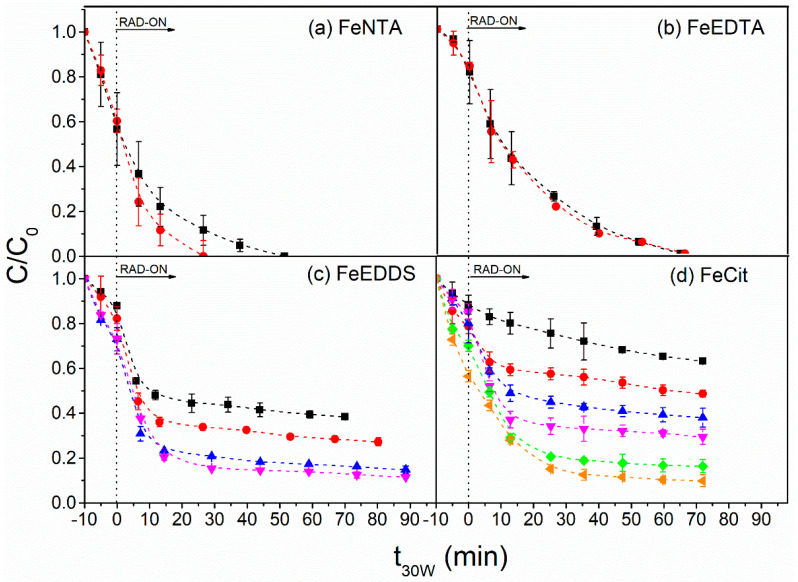
Influence of Fe/L molar ratio (■ 1:1, ● 1:2, ▲ 1:3, ▼ 1:4, ♦ 1:5 and ◄ 1:6) on the degradation of the mixture of benzophenone-3, fipronil and propylparaben using the iron complexes (**a**) FeNTA, (**b**) FeEDTA, (**c**) FeEDDS and (**d**) FeCit. Initial conditions: [CEC] = 100 µg L^−1^ (for each compound), [H_2_O_2_] = 5.9 mmol L^−1^, [Fe^3+^] = 100 µmol L^−1^; pH = 6.9 (the natural pH of the MWWTP effluent).

**Figure 3 molecules-27-05521-f003:**
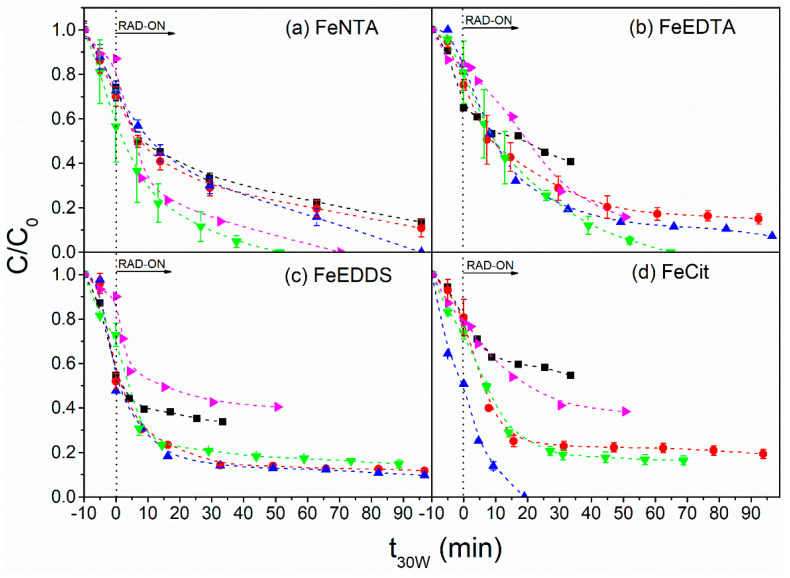
Influence of H_2_O_2_ concentration (■ 0.74 mmol L^−1^, ● 1.5 mmol L^−1^, ▲ 2.9 mmol L^−1^, ▼ 5.9 mmol L^−1^ and ► 11.8 mmol L^−1^) on the mixture degradation of benzophenone-3, fipronil and propylparaben using the iron complexes (**a**) FeNTA, (**b**) FeEDTA, (**c**) FeEDDS and (**d**) FeCit. Initial conditions: [CEC] = 100 µg L^−1^ (for each compound), Fe:L = 1:1 (FeEDTA and FeNTA); 1:3 (FeEDDS) and 1:5 (FeCit), [Fe^3+^] = 100 µmol L^−1^; pH = 6.9 (the natural pH of the MWWTP effluent).

**Figure 4 molecules-27-05521-f004:**
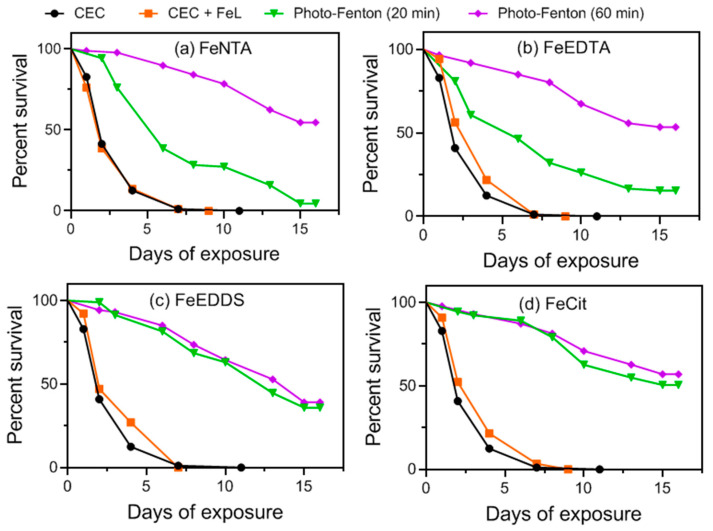
*D. melanogaster* survival assay for MWWTP effluent samples spiked with the mixture of benzophenone-3, fipronil and propylparaben before and after solar/photo-Fenton treatment using different iron complexes: (**a**) FeNTA, (**b**) FeEDTA, (**c**) FeEDDS and (**d**) FeCit. Initial conditions: [CEC] = 100 µg L^−1^ (for each compound), Fe/L = 1:1 (FeEDTA and FeNTA); 1:3 (FeEDDS) and 1:5 (FeCit), [Fe^3+^] = 100 µmol L^−1^; [H_2_O_2_] = 1.5 mmol L^−1^ (for FeEDDS), 2.9 mmol L^−1^ (for FeCit) and 5.9 mmol L^−1^ (for FeEDTA and FeNTA).

**Table 1 molecules-27-05521-t001:** Pseudo-first kinetic constants and cost assessment for solar photo-Fenton using different iron complexes.

Parameter	FeCit	FeEDDS	FeEDTA	FeNTA
k (min^−1^)	0.10	0.036	0.051	0.063
R^2^	0.96	0.97	0.99	0.99
t_1/2_ (min)	6.6	19	14	11
Cost_Iron_ (USD m^−3^)	5.6	5.6	5.6	5.6
Cost_Ligand_ (USD m^−3^)	13	321	7.7	4.5
Cost_Oxidant_ (USD m^−3^)	1.4	0.72	2.8	2.8
Cost_Total_ (USD m^−3^)	20	327	16	13

## Data Availability

Not applicable.

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
