# Peer review of "Effect of Iron Complex Source on MWWTP Effluent Treatment by Solar Photo-Fenton: Micropollutant Degradation, Toxicity Removal and Operating Costs"

_molecules, 2022, doi:10.3390/molecules27175521_

Round 1

Reviewer 1 Report

The present work explains the efficiency of solar photo-Fenton using four iron ligands for simultaneous degradation of benzophenone-3, fipronil and propylparaben at municipal waste water treatment plant effluent. The results discussed in the present study are quite interesting. I recommend the publication of the article with few minor corrections/ explanations.

Authors have explained the experimental work and have presented the results in a comprehensive and detailed manner. Authors have very clearly and explicitly explained the reactions and mechanisms occurring in the process. In all, it’s a very well written article. The only issue with the current research article is with its figures. I feel the figures should b more explanatory. There should be of better quality. Moreover, x-axis is missing for fig. 1 (a) and for fig.1 (b). Similar issues are observed for Fig. 2 and Fig. 3 as well. In fig. 1, fig. 2 and fig. 3 how the data is recorded before time t=0. Author may please explain.

Reviewer 2 Report

The presented manuscript includes the study of the effect of iron complex source on MWWTP effluent treatment by solar photo-fenton: micropollutant degradation, toxicity removal and operating costs. The paper is of interest.

The results of the work are presented on a good level, well-structured and well written. References are updated. Results compared with published analogs. Minimal requirements for statistical analysis were done.

Reviewer 3 Report

This paper describes the effect of the iron complex on solar photo-Fenton degradation of micropollutants. The paper is well written.

1.        The authors should revise the abstract and conclusion parts for highlighting the novelty of their research.

2.        The introduction should be revised by the addition of previous works done on the same reaction using other or the same materials. The authors should describe the shortcomings of the already reported similar works in literature and how the proposed research can bring changes.

3.        Figures are blur. Figures in the manuscript should be revised with more clear and high resolutions ones.

4.        The authors should replace the older papers in references with the recently published papers to make their research more up to date. Such as

https://doi.org/10.1007/s13762-022-04216-6

https://doi.org/10.3390/ijerph16112066
